# The Connection between Phuket's Water Supply and the Hotel Sector's Water Use for Assessment of Tourism Carrying Capacity

Thunyapat Sattraburut , Kritana Prueksakorn * , Thanchanok Kitcharoen, Teeraporn Amattayakul, Papaporn Pinitsuwan and Chitsanuphong Pratum *

Faculty of Environment and Resource Studies, Mahidol University, Nakhon Pathom 73170, Thailand; thunyapat.sat@mahidol.ac.th (T.S.); thanchanok.kitcharoen@gmail.com (T.K.); nextteeraporn@gmail.com (T.A.); aor89890@gmail.com (P.P.)

* Correspondence: kritana.pru@mahidol.ac.th (K.P.); chitsanuphong.pra@mahidol.ac.th (C.P.)

**Abstract:** For tourism development in areas where there are frequent problems with water shortage, it is important to assess water use potential from a geographic perspective. This study investigated the relationship between water use in the hotel sector and the amount of available water in Phuket for tourism carrying capacity assessment. Geographic information system (GIS) was applied to support spatial analysis. The studied hotels were in the size required to conduct an environmental impact assessment (EIA), totaling 178 hotels. There would be a total water use of 24,275 m$^3$/day and 40,457 m$^3$/day in the low and high seasons, respectively. In terms of annual water use, in the cases of lowest, normal, and highest possible consumption, there would be water consumption amounts of 8,860,021 m$^3$/year, 11,303,606 m$^3$/year, and 14,766,699 m$^3$/year, respectively. From evaluating the capacity to support tourists in terms of water adequacy in Phuket Province, our results reveal that the amount of water in the province is insufficient to meet the needs of tourists. This is because the number of tourists coming to travel each year is ordinarily more than 10 million people, and there is a plan to accommodate 12 million tourists in 2023, but the amount of water that the province can procure is sufficient to support a maximum of not more than 8,986,600 tourists per year only if the water that EIA hotels store and produce themselves is included. This amount of water is still insufficient for the targeted number of tourists and other sectors, and the province's water demand is likely to increase. In this regard, Phuket should hastily prepare a plan and measures to increase the amount of available water within the province.

**Keywords:** environmental impact assessment; hotel; Phuket; tourism carrying capacity; water demand; water supply

## 1. Introduction

Water is an essential factor in human life, serving various purposes such as consumption, agriculture, industry, and tourism. These activities require a relatively high amount of water [1–6]. Phuket is one of the most popular tourist destinations in Thailand, known for its beautiful beaches, crystal-clear waters, food, and lively nightlife. Consequently, it attracts a large number of both domestic and international tourists each year. The tourism industry can, therefore, be regarded as a crucial driver of the local economy [7–9], resulting in water-dependent activities in many sectors, including households, hotels, restaurants, entertainment venues, and natural attractions [10,11].

The Project Management Office, Royal Irrigation Department [12] analyzed historical data on water demand from all activities in Phuket. It was found that the demand for water in different areas tends to increase continuously, averaging about 1.28% per year. This finding aligns with data from the Royal Irrigation Department, indicating that in 2021, Phuket used approximately 77 million m$^3$ of water to support economic growth and tourism. Specifically, water consumption for tourism reached 20 million m$^3$ [13].

Furthermore, forecasted data for 2027 and 2040 reveals that Phuket's water demand will rise to 87.67 million m³ and 104.93 million m³, respectively, with a substantial volume of 23.58 million m³ and 33.18 million m³ of water used for tourism alone [14].

Phuket's tourism industry is largely driven by the hotel sector. Generally, the hotel's primary water source is tap water from the Provincial Waterworks Authority. Other than that, some hotels utilize private waterworks, while the remainder relies on self-drilled groundwater [15]. Nonetheless, as is the case with many tourist cities worldwide, Phuket faces similar challenges, particularly severe water shortages during the dry season (January and February) each year [16]. The hotel industry is currently recovering from an increase in tourist arrivals following the end of the 2019 coronavirus outbreak, as reported by the World Health Organization [17]. However, it is anticipated that rainfall and runoff will begin to decrease due to the El Niño phenomenon [18–20]. Phuket is about to face a more serious water shortage, and it is difficult to solve the problem because the area is an island and has limited water storage [21].

When considering the tourism carrying capacity (in terms of the number of rooms and available water alone), it becomes evident that available water often limits the number of tourists in Phuket. Several studies have reported that the number of tourists is directly correlated to the number of hotels. Therefore, hotels are considered to be the main water users [22–25]. Moreover, the water consumption of hotels is correlated to the number of rooms [26–28], beds [25,29], room area size [30], and the number of nights stayed [31]. While studies in Phuket have assessed the water demand of hotels and the factors that affect the amount of water used, there has been no study of the connection between water use in the hotel sector and spatial water availability, as well as no evaluation of the tourism carrying capacity in terms of water adequacy in the areas where those hotels are located.

Thus, this research aims to study the relationship between water consumption in the hotel sector and the availability of water in Phuket. Additionally, water adequacy has been assessed for other sectors, particularly in the context of normalizing tourism, to study the appropriate carrying capacity of Phuket Province. Furthermore, guidelines for augmenting the water supply in Phuket Province are also proposed.

## 2. Materials and Methods

### 2.1. Study Area

Phuket is a province in Thailand; it is also the name of the largest island in the country. It is situated in the upper southern region of Thailand, positioned between latitude 7°45′ to 8°15′ N and longitude 98°15′ to 98°40′ E (Figure 1). It comprises Phuket Island, along with an additional 32 smaller islands located along its coastline.

This study focuses on the main island of Phuket, which comprises a total area of 543 km² out of 576 km², accounting for 94.28% of the province's total area. It is divided into 70% mountainous areas and approximately 30% flat regions in the central and eastern parts of the island. The western coastal area is characterized by mountains and sandy beaches, while the eastern coast features mudflats and mangrove forests. In terms of climate, Phuket experiences an equatorial climate that is hot and humid year-round due to the influence of the Southwest and Northeast monsoons. There are two distinct seasons in Phuket: the rainy season from April to November and the dry season from December to March [32].

Phuket Province consists of three districts: Mueang Phuket District, Kathu District, and Thalang District. As of 2021, the population of Phuket stands at 414,471 people, with Mueang District having the highest number of residents, followed by Thalang and Kathu Districts, respectively. Phuket features 24 small watersheds scattered across the province, covering a catchment area of 1244 km², along with nine major canals. Additionally, there is a source of surface water from peatland, encompassing an approximate area of 0.912 km². The storage of 12,022,500 m³ of water has also been facilitated by abandoned mines in Phuket. Furthermore, groundwater from Thep Kasattri Subdistrict, Thalang District, can be harnessed for consumption. It is located at a depth of 20–40 m, with the water volume falling within the range of 10–30 m³ per hour [33].

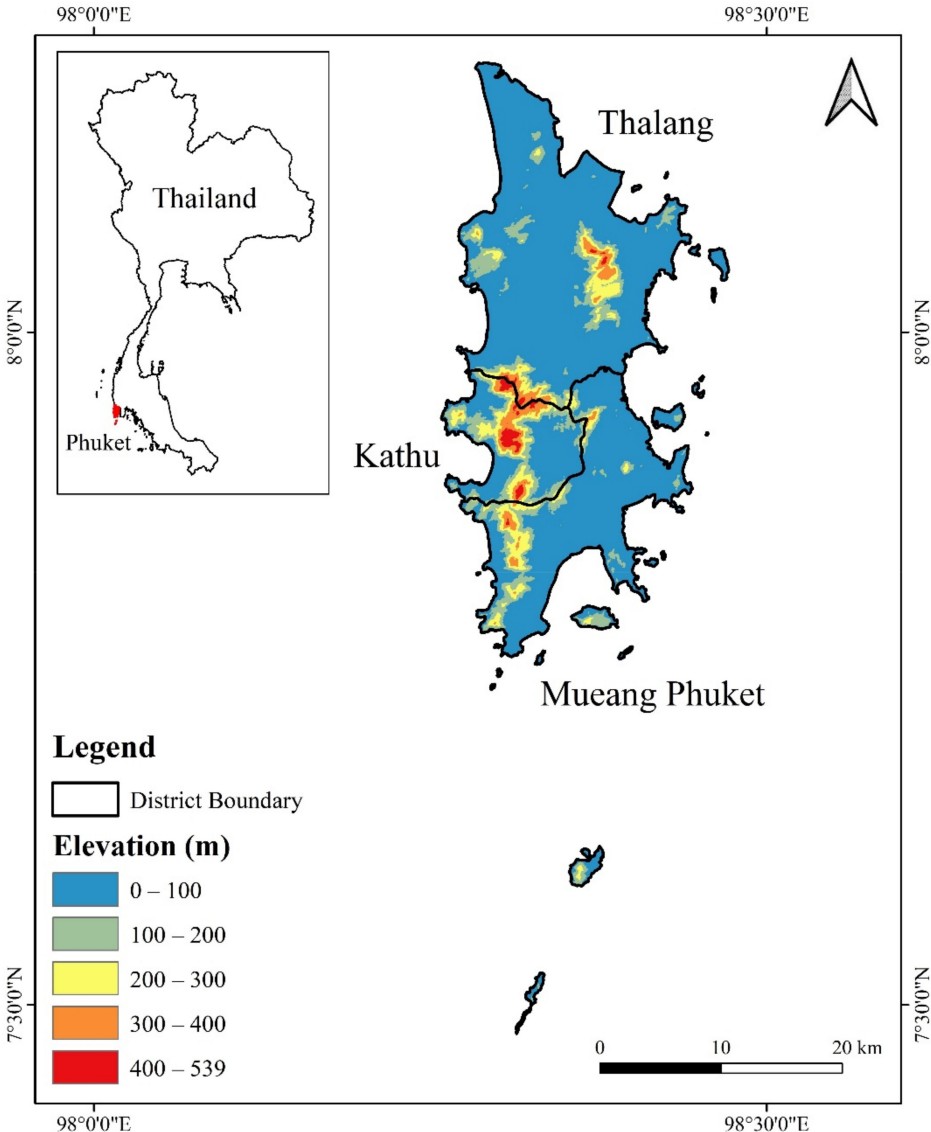

**Figure 1.** Location of Phuket, Thailand, with an elevation.

*2.2. Data Collection*

This research was a quantitative study that focused on water use in the hotel sector. Specifically, it examined hotels that are required to conduct an environmental impact assessment (EIA) within Phuket, referred as the "EIA hotel" in this study. Due to the outbreak of Coronavirus 2019, staying at hotels in Phuket during the years 2019 to 2022 cannot be considered normal. Therefore, the research team decided to analyze data from 2017 instead. The data gathered for this research in 2017 is categorized into several aspects: water consumption, water sources used in the hotel sector, amount of water supply from the Provincial Waterworks Authority, and water volume data from private water reservoirs.

The collected data consist of secondary data obtained from research papers, government agencies, and academic institutions. The conceptual framework of the study is illustrated in Figure 2, and the specifics of the retrieved information and reference sources are presented in Table 1.

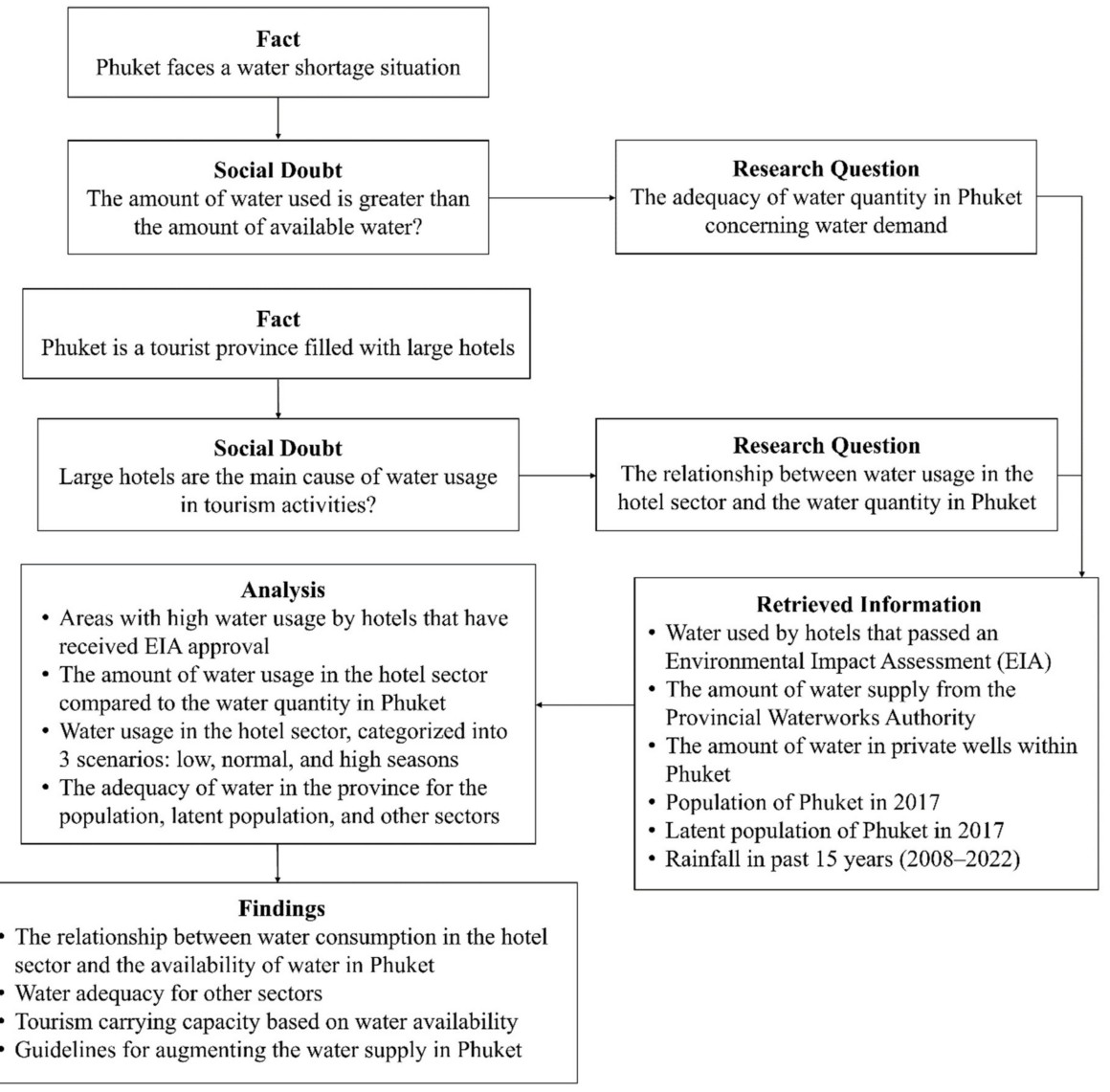

**Figure 2.** Conceptual framework of the study.

**Table 1.** Retrieved information and reference sources.

| Retrieved Information | Type of Data | Reference Sources |
|---|---|---|
| Water used by hotels that passed an Environmental Impact Assessment (EIA) | -number of rooms<br>-total water consumption (m³/day)<br>-water sources | Environmental Impact Assessment Information Center Office of Natural Resources and Environmental Policy and Planning [34] |
| The amount of water supply from the Provincial Waterworks Authority | -amount of water produced<br>-amount of water distributed<br>-amount of water sold | Provincial Waterworks Authority, Phuket Branch [35] |
| The amount of water in private wells within Phuket | -amount of water | Performance report from Phuket Water Management Research Project [36] |
| Population of Phuket in 2017 | -number of people | Phuket Provincial Office, Strategy and Information Group for Development of Phuket Province [32] |
| Latent population of Phuket in 2017 | -number of latent people | National Statistical Office, Ministry of Digital Economy and Society [37] |
| Rainfall in past 15 years (2008–2022) | -monthly rainfall | Meteorological Department [38] |

The study was divided into the following sections:

(1) Analysis of water quantity in Phuket Province that can be calculated from the following equation:

$$Xt = Xa + Xb + Xc \qquad (1)$$

where Xt = Total water amount in Phuket province ($m^3$/year)
Xa = water amount from the Provincial Waterworks Authority, Phuket branch ($m^3$/year)
Xb = water amount from private water reservoirs ($m^3$/year)
Xc = water amount that the hotels stored and produced themselves ($m^3$/year)

(2) Analysis of the water adequacy for use in the province for the population, latent population, and other sectors.

(3) Analysis of the water amount used by the hotels, divided into (1) the water amount used by EIA hotels and is currently in operation, and (2) the water amount used by EIA hotels but are not operating (there is potential for future operation).

(4) Simulation of water situations to predict the amount of water used during and outside the tourist season, divided into two cases: (1) calculation of daily water use with the assumption that a high season scenario accounts for 100% and a low season scenario accounts for 60% of water use in each hotel (according to [36]) and (2) calculation of annual water use by assuming three situations, divided into a low case scenario accounting for 60% of water use in hotels, a high case scenario accounting for 100% of water use in hotels, and a normal case scenario accounting for water use according to season.

(5) Analysis of the proportion between the amount of water used in the hotel sector and the amount of water in Phuket Province, divided into two cases: (1) Hotels do not have the potential to store and produce their own water, and (2) Some hotels have the potential to store and produce their own water.

(6) Assessment of the tourism carrying capacity, calculated from the following equations (adapted from [21,39]):

$$Wc = (A \times Wa) + (B \times Wb) \qquad (2)$$

where Wc = total water consumption (L/day)
A = number of population according to civil registration (person)
B = latent population (person)
Wa = average water use rate of the population according to the civil registration (L/person/day)
Wb = average water use rate of the latent population (L/person/day)

$$TCCw = (Ws - Wc)/Wt \qquad (3)$$

where TCCw = tourism carrying capacity (person)
Ws = amount of water used in the province (L/day)
Wc = total water use (L/day)
Wt = average water use rate of tourists (L/person/day)

(7) Analysis of rainfall data from the Meteorological Department as a basis to evaluate the potential for increasing the amount of water availability in Phuket Province.

## 3. Results and Discussion

### 3.1. The Quantity of Water in Phuket Province

The majority of hotels in Phuket primarily use three major sources of water: The Provincial Waterworks Authority, private companies/private water wells and reservoirs, and the water source that the hotels collect and produce on their own (such as the drilling of artesian wells within the hotel's areas). Consequently, the amount of water in Phuket is taken into consideration when determining the sum of the water quantities obtained from these three sources (Equation (1)). This total is then compared with the amount of water

used in the hotel sector and other sectors to assess the proportion of water usage and the adequacy of water supply to meet the province's water demand.

In 2017, the water supply from the Provincial Waterworks Authority in Phuket reached 22,369,452 m$^3$, as reported by [40]. As for the amount of water in private water reservoirs, the total amount of water is 14,099,006 m$^3$/year [36], and there could be 4,000,000 m$^3$/year of groundwater [40]. However, there was no publicly available information regarding water sources collected and produced by hotels themselves, such as the drilling of artesian wells in hotel areas. Therefore, the researchers obtained data from the EIA reports, which indicated that hotels collected a total of 2,587,704 m$^3$/year. As a result, when combining these main sources of water, the estimated total water availability in Phuket is 43,056,162 m$^3$/year. When considering the distribution of water sources, it was found that the Provincial Waterworks Authority contributed 52.0%, private companies/private wells and reservoirs accounted for 42.0%, and water sources collected and produced by hotels themselves constituted 6.0% of the total water supply.

*3.2. The Adequacy of Water in Phuket for the Population, Latent Population, and Other Hotels If EIA Hotels Are Prioritized*

If all EIA hotels, which use a total of 14,766,699 m$^3$/year, are given priority before other groups of water users, there is a remaining water supply of 24,289,463 m$^3$/year available for use in the province (without considering the amount of groundwater). Groundwater is considered a main water source distributed to the residents and other sectors of Phuket. In 2010, the total groundwater supply was approximately 4 million m$^3$/year [41]. Therefore, the annual water availability for the province is 28,289,463 m$^3$ (Figure 3).

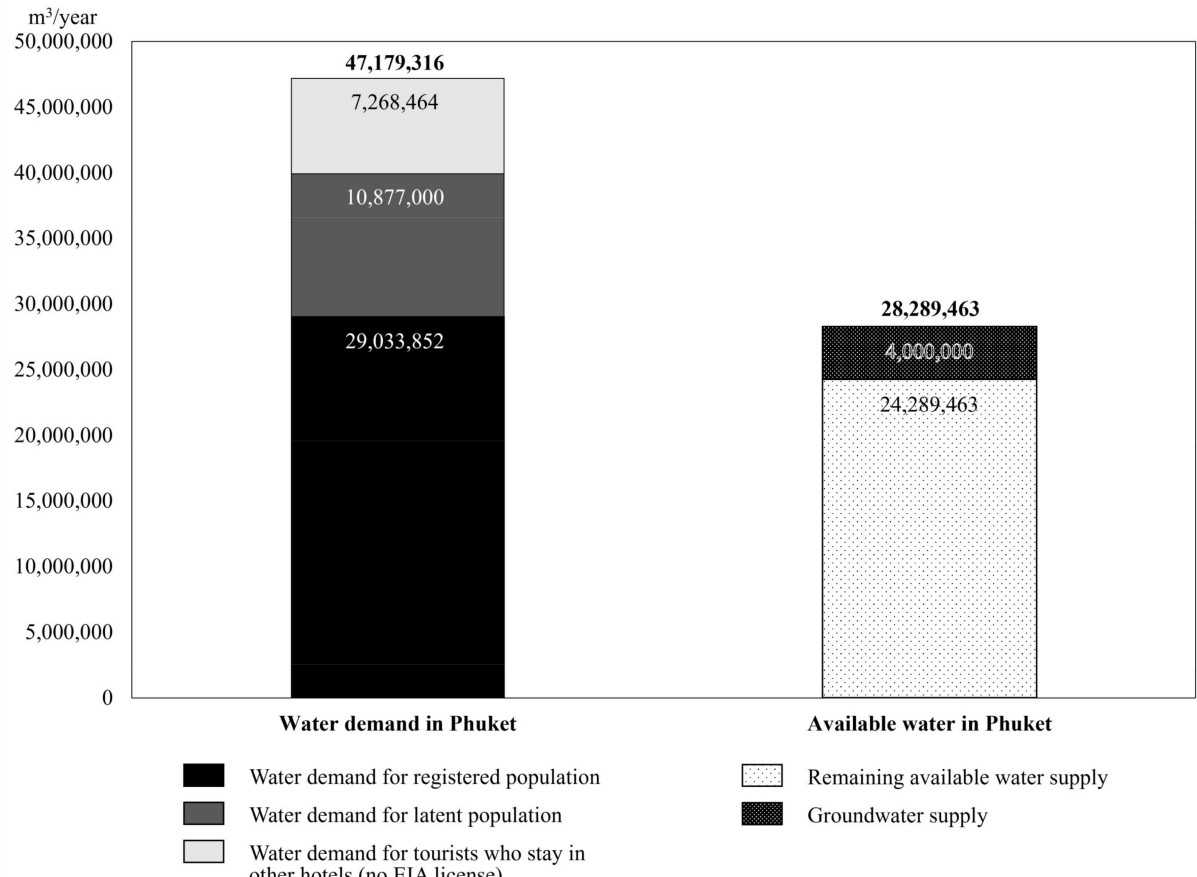

**Figure 3.** The adequacy of water in the province for the population, latent population, and tourists who stay in other hotels if EIA hotels are prioritized.

For water use by Phuket citizens, the analysis of water adequacy in the province revealed that the water usage rate varies among the administrative districts of Phuket Province. This variation is attributed to the water usage patterns of local residents and their access to public utilities [41]. The determination of water usage levels based on the population in each administrative area, as well as the calculation of the average usage rate for residents in Phuket Province, are presented in Tables 2 and 3. Table 2 also shows that the rise in water consumption is directly correlated with population size. The areas with higher population densities generally exhibit more water-related activities than areas with lower population densities.

**Table 2.** The water use rate consumption according to the population in the administrative area [41].

| Population (Person) | Water Use Rate (L/Person/Day) |
| --- | --- |
| 3000–10,000 | 120 |
| 10,001–20,000 | 170 |
| 20,001–30,001 | 200 |
| 30,001–50,000 | 250 |
| >50,000 | 300 |

**Table 3.** The average water usage rate for the population in Phuket Province [32].

| District | Population [1] | Water Use Rate (L/Person/Day) |
| --- | --- | --- |
| Mueang Phuket | | |
|   Phuket Metropolitan Municipality | 79,252 | 300 |
|   Karon Subdistrict Municipality | 8128 | 120 |
|   Ratsada Subdistrict Municipality | 46,955 | 250 |
|   Wichit Subdistrict Municipality | 49,187 | 250 |
|   Rawai Subdistrict Municipality | 18,052 | 170 |
|   Outside the municipality | 39,259 | 250 |
| Kathu | | |
|   Patong Metropolitan Municipality | 20,721 | 200 |
|   Kathu Metropolitan Municipality | 28,674 | 200 |
|   Outside the municipality | 6949 | 120 |
| Thalang | | |
|   Choeng Thale Subdistrict Municipality | 6938 | 120 |
|   Thep Kasattri Subdistrict Municipality | 8316 | 120 |
|   Outside the municipality | 85,293 | 300 |
| Total | 397,724 | Average = 200 |

[1] Information presented in 2017.

In 2017, the population of Phuket was 397,724 people, and the average water usage rate per person for that year was 200 L/person/day. Therefore, the water demand from the population was approximately 79,545 $m^3$/day or 29,033,852 $m^3$/year. The assessment of water adequacy in the province for the population in this study revealed that the remaining water volume was 28,289,463 $m^3$. This amount can be distributed to approximately 387,526 people, which accounts for 97.4% of the total provincial population. The available water in the province does not appear to be sufficient for the entire population, as indicated above. It is necessary to provide water from other sources for the remaining 10,189 people, constituting 2.6% of the total population of the province.

Additionally, Phuket is also the province with the 9th largest latent population in Thailand. According to [37], Phuket has a latent population of 149,000 people. The latent population refers to people who live in a province but whose names do not appear in the house registration within the province in which they reside. Therefore, the average water usage rate is set to be equal to the population in the province, which is 200 L/person/day. The results of this study show that water demand was 29,800 $m^3$/day or 10,877,000 $m^3$/year for these populations.

Information from the Ministry of Tourism and Sports [42] reveals that the hotel sector in the province has a total of 92,267 rooms. The number of hotel rooms that do not require EIA was up to 56,896. When calculating the water usage rate of tourists staying at 350 L/person/day [41], with a limit of 1 tourist per 1 room, the total water consumption needs for these hotels amount to 19,914 m$^3$/day or 7,268,464 m$^3$/year, as an extreme case of estimation. When considering the water demand of (1) the registered population, (2) the latent population, and (3) the hotels that do not require EIA, it becomes evident that Phuket Province still needs to procure at least 18,889,853 m$^3$ of water per year. Furthermore, there have been changes in global or regional climate patterns, particularly notable over the past three years (from September 2020 to March 2023). The world has experienced an El Niño phenomenon, leading to reduced rainfall in Southeast Asia and Australia, resulting in more severe and prolonged droughts [43–45]. The UK Weather Bureau also predicts that temperatures in 2023 will be higher than those in 2022 [46]. As a result, Thailand is grappling with increasingly severe weather conditions characterized by lower-than-usual rainfall, drought problems, and water shortages in many areas [46]. In 2023, the Tourism Authority of Thailand anticipates that the number of foreign tourists visiting Thailand will exceed 18 million due to the improving situation concerning the spread of the coronavirus (COVID-19) [47]. Given the aforementioned circumstances, Phuket Province may also confront more acute water shortages. Therefore, the most suitable course of action is to plan for an augmentation of the water supply in the province to address potential future water shortages.

*3.3. Water Consumption of Hotels Requiring an Environmental Impact Assessment (EIA)*

This section of the study emphasizes the patterns and water consumption of EIA hotels located in Phuket Province because such hotels use a lot of water, which affects the overall water use in Phuket Province. Among a total of 178 hotels mandated to undergo an EIA, 166 hotels were operating in areas requiring EIA preparation, and these hotels were further categorized as follows: 71 in Mueang District, 60 in Kathu District, and 35 in Thalang District. Moreover, there were an additional 12 EIA hotels required to have EIAs and had the potential to open for future operations. These hotels fell into three distinct types: (1) hotels with temporary or permanent closure, (2) hotels with EIA approval and currently under construction, and (3) hotels with EIA approval but not currently under construction. Of these hotels, two were located in Mueang District, and ten were situated in Thalang District (Figure 4).

The three main water sources in the studied hotels are categorized as follows: The Provincial Waterworks Authority, private companies/private water wells and reservoirs, and the source of water collected and produced by the hotels themselves (such as the drilling of artesian wells within hotel premises). Rainwater serves as the main source of water for producing tap water. Each of the studied hotels has different main water sources, depending on their location and water consumption. Table 4 provides information on water sources for hotels that are already operating and potential hotel sites to be opened in the future. Some hotels use water from only one source, while other hotels use water from two or more sources. For example, two hotels that are already in operation and one hotel with the potential to be opened in the future use water from The Provincial Waterworks Authority and private companies.

The water consumption of all 178 hotels was 14,766,699 m$^3$/year. The total water consumption consists of three parts: (1) water consumption from the room, (2) water consumption in the common areas of the hotel, e.g., shared kitchens, shared bathrooms, swimming pools, and gymnasiums, and (3) water consumption from other parts relating to a hotel, but that is not directly related to guests such as bathrooms, dining rooms, and kitchens.

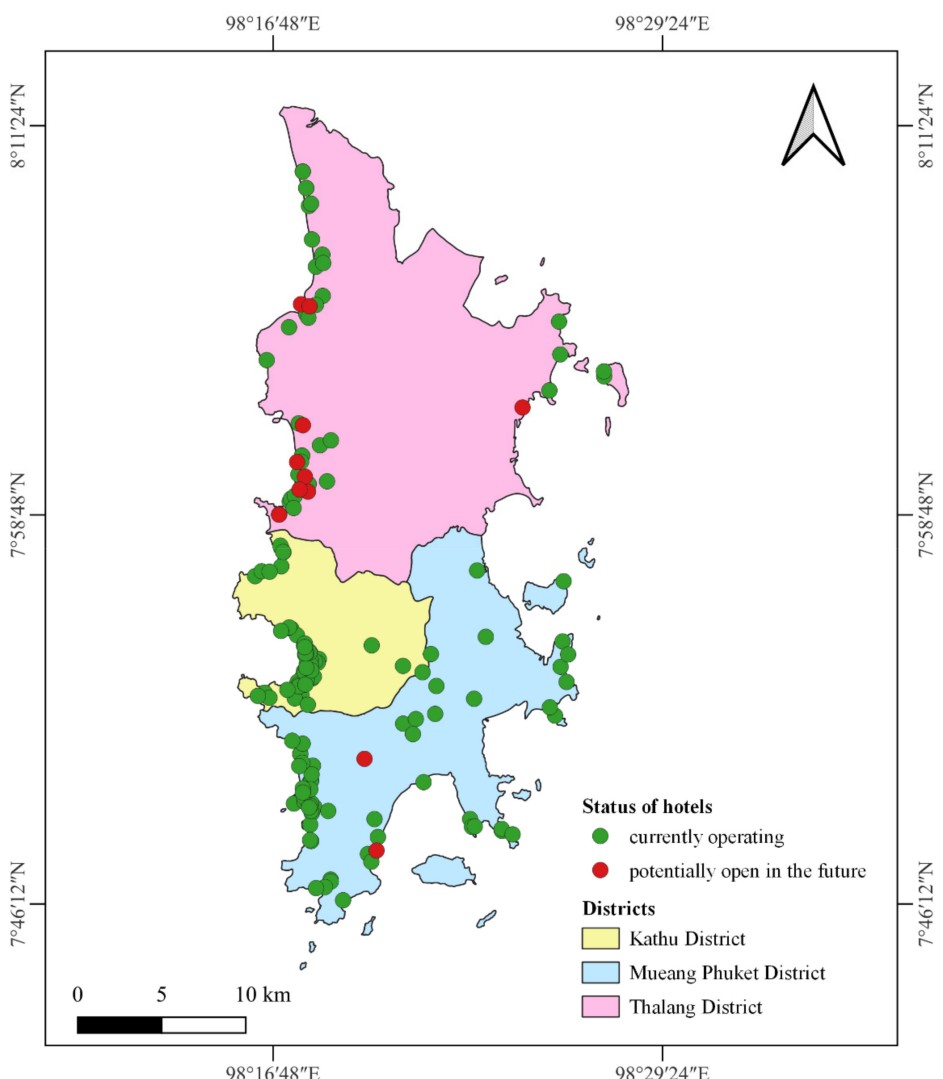

**Figure 4.** The locations of the hotels requiring an Environmental Impact Assessment (EIA) in Phuket.

**Table 4.** The major water sources for the studied hotels.

| Sources of Water Supply for the Studied Hotels | Number of Hotels |
|---|---|
| Hotels that are already in operation | |
| 1. The Provincial Waterworks Authority | 95 |
| 2. Private companies/private water wells and reservoirs | 25 |
| 3. The hotels storing and producing their own water | 31 |
| 4. The Provincial Waterworks Authority and private companies | 2 |
| 5. The Provincial Waterworks Authority and the hotels storing and producing their own water | 6 |
| 6. Private companies and the hotels storing and producing their own water | 6 |
| 7. The Provincial Waterworks Authority, private companies, and the hotel storing and producing its own water | 1 |
| Total | 166 |
| Hotels with a potential to be opened in the future | |
| 1. The Provincial Waterworks Authority | 7 |
| 2. Private companies/private water wells and reservoirs | 1 |
| 3. The hotels storing and producing their own water | 3 |
| 4. The Provincial Waterworks Authority and private companies | 1 |
| Total | 12 |

### 3.3.1. The Water Consumption of the Hotels That Requires an Environmental Impact Assessment (EIA) and Are Currently Operating

The research carried out in this section has revealed that the 166 hotels collectively use 14,291,885 tons of water per year. The water consumption of each hotel depends largely on factors such as size, the number of rooms, and the activities conducted within the scope of the project, including swimming pools, kitchens, dining areas, etc.

According to Figure 5, it was observed that the western part of Phuket experiences higher water consumption compared to other areas. This can be attributed to the hotels' locations, given that most of the studied hotels were situated in the western part of the province. In the case of Mueang District, hotels are not only located in the western area but also sprawl across the entire district. Additionally, the map provides a breakdown of water consumption (m$^3$/day), categorized into six ranges, as outlined in Table 5.

The study revealed an interesting range of 1251 m$^3$/day to 1500 m$^3$/day for two hotels that used more water compared to others. Several factors influence the water consumption of these two hotels in comparison to other hotels, including the project area, room types, the number of rooms, and the activities within the hotel (e.g., swimming pools, water parks, and golf courses). This explains why larger hotels with more rooms and a wider range of on-site activities tend to have higher water consumption per room than smaller hotels.

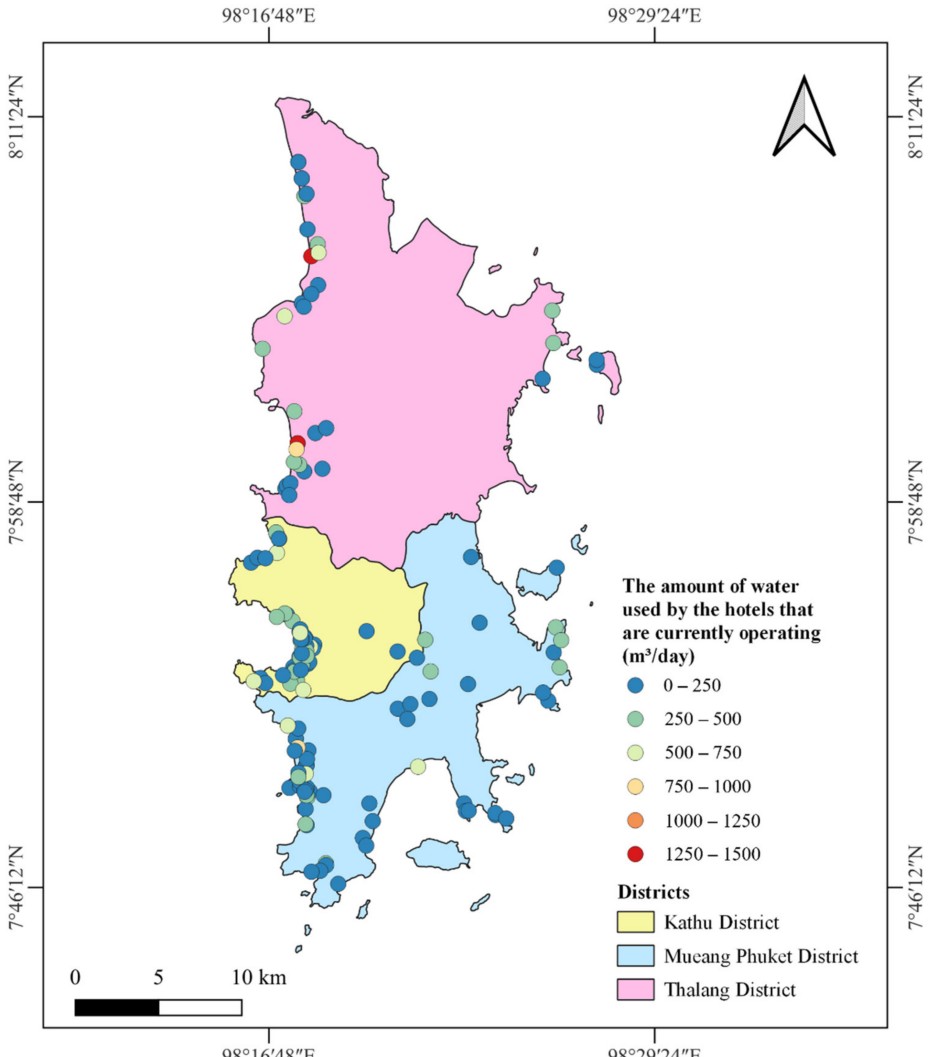

**Figure 5.** The locations of the operating hotels that requires an Environmental Impact Assessment (EIA) and their water consumption per day.

**Table 5.** The water consumption range and the number of hotels in each range.

| The Water Consumption Range (m³/Day) | The Number of Hotels in Each Water Consumption Range (Hotels) |
|---|---|
| 0–250 | 113 |
| 251–500 | 38 |
| 501–750 | 11 |
| 751–1000 | 2 |
| 1001–1250 | 0 |
| 1251–1500 | 2 |

This research also compares the water consumption of hotels with previous research conducted in Phuket [15], which analyzed the water consumption of eight (medium and large) hotels. The overall water consumption in both studies was found to be similar, with an average difference of approximately 1621 m³/month. However, the total water consumption is higher in this study, totaling 40,831 m³/month, primarily due to the presence of slightly more rooms. Therefore, both studies demonstrate the same trend regarding water consumption. The details of the studies in this section are presented in Table 6.

**Table 6.** The water consumption of eight hotels in the research conducted in 2013 [15] compared to the water consumption of eight hotels in this research (Research team, 2023).

| Comparison | Results from [15] | Result from This Study |
|---|---|---|
| The number of rooms (room) | 1244 | 1352 |
| The hotel size | 5 medium-sized hotels [1] and 3 large-sized hotels [2] | 5 medium-sized hotels and 3 large-sized hotels |
| Total water consumption at 8 locations (m³/day) | 1307 | 1361 |
| Total water consumption at 8 locations (m³/month) | 39,210 | 40,831 |

[1] Medium-sized hotels: hotels with 50–150 rooms. [2] Large-sized hotels: hotels with more than 150 rooms.

The analysis of average water consumption per day of the operating EIA hotels is presented as follows:

(1) Average water usage per room per day divided by hotel size (small, medium, and large) calculated from the total water usage of the hotel (including administrative activities and all other parts).

As shown in Figure 6, there are a total of 166 operating EIA hotels (with 34,032 rooms as per Table 7), and their average water usage is $1.1 \pm 0.6$ m³/room/day. A large hotel has the highest water usage at 5.5 m³/room/day with a total of 239 rooms, a golf course, a communal swimming pool, and a swimming pool for an individual room. Meanwhile, a hotel with the least water usage is at 0.3 m³/room/day. It is also a large hotel with a total of 219 rooms.

From calculating the average water use per room per day divided by hotel size as per Table 8, the average water use of a medium-sized hotel has the highest value at $1.2 \pm 0.7$ m³/room/day while the average water uses of large and small hotels have the values at $1.1 \pm 0.6$ and $1.0 \pm 0.3$ m³/room/day, respectively. It can be seen that the average water usage of large hotels has a higher SD than small hotels and has the highest range from 0.3–5.5 m³/room/day, which can be assumed that large hotels have the highest water usage due to the variety of activities to accommodate tourists. Meanwhile, many large hotels have effective water management systems (water recycling and circulating for internal use in various hotel activities).

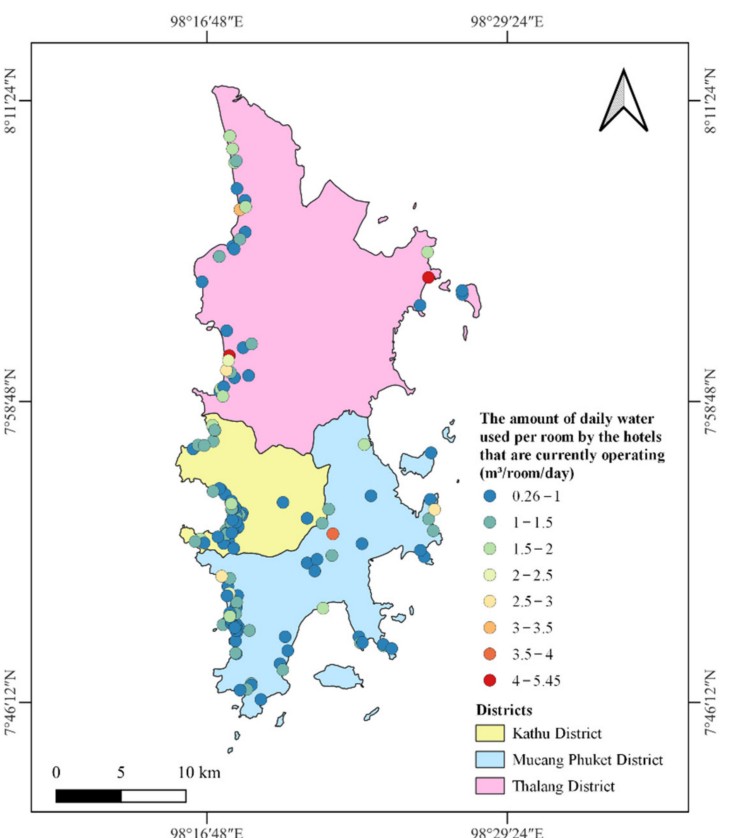

**Figure 6.** The locations of the operating hotels that require an Environmental Impact Assessment (EIA) and their water consumption per room per day.

**Table 7.** Average water usage per room per day, calculated from the operating hotels' activities.

| Daily Water Consumption per Room (m³/Room/Day) | Number of Hotels | Total Rooms |
|:---:|:---:|:---:|
| <1.0 | 99 | 19,851 |
| 1.0–1.5 | 42 | 8784 |
| 1.5–2.0 | 14 | 3038 |
| 2.0–2.5 | 4 | 981 |
| 2.5–3.0 | 3 | 478 |
| 3.0–3.5 | 1 | 452 |
| 3.5–4.0 | 1 | 120 |
| >4.0 | 2 | 328 |
| Total | 166 | 34,032 |

**Table 8.** Average water usage per room per day, divided by the operating hotels' sizes.

| Hotel Size | Number of Hotels | Total Rooms | Total Water Consumption (m³/Day) | Water Use per Room (m³/Room/Day) | | | |
|:---|:---:|:---:|:---:|:---:|:---:|:---:|:---:|
| | | | | Average | SD | Max | Min |
| Small [1] | 4 | 151 | 154.9 | 1.0 | 0.3 | 1.4 | 0.8 |
| Medium [2] | 53 | 5654 | 6888.2 | 1.2 | 0.7 | 4.1 | 0.4 |
| Large [3] | 109 | 28,227 | 32,112.8 | 1.1 | 0.6 | 5.5 | 0.3 |
| Total | 166 | 34,032 | 39,155.86 | | | | |

[1] Small-sized hotels: hotels with less than 50 rooms. [2] Medium-sized hotels: hotels with 50–150 rooms. [3] Large-size hotels: hotels with more than 150 rooms.

(2) Average water usage per room per day divided by hotel location

According to Table 9, the average water use of hotels located in Thalang District had the highest value at $1.5 \pm 1.0$ m$^3$/room/day. This is quite a noticeable difference compared to hotels located in Mueang District and Kathu District, with water usage at $1.1 \pm 0.5$ m$^3$/room/day and $1.0 \pm 0.3$ m$^3$/room/day, respectively. Figure 6 shows that the hotels with high water usage are mostly located in the southwestern area of Thalang District near West Beach.

**Table 9.** Average water usage per room per day, divided by the districts where the hotels are located.

| District | Number of Hotels | Total Rooms | Total Water Consumption (m$^3$/Day) | Water Use per Room (m$^3$/Room/Day) | | | |
|---|---|---|---|---|---|---|---|
| | | | | Ave | SD | Max | Min |
| Kathu | 59 | 14,250 | 14,469.5 | 1.0 | 0.3 | 1.8 | 0.6 |
| Mueang | 71 | 12,831 | 13,964.5 | 1.1 | 0.5 | 3.7 | 0.4 |
| Thalang | 36 | 6951 | 10,722.0 | 1.5 | 1.0 | 5.5 | 0.3 |
| Total | 166 | 34,032 | 39,155.86 | | | | |

### 3.3.2. The Amount of Water Used by the Hotels That Require an Environmental Impact Assessment (EIA) and Have the Potential to Open in the Future

There are 12 EIA hotels with the potential to open for business in the future. They have a total water consumption of 474,814 m$^3$/year, with the hotel's water consumption being less than 250 m$^3$/day (Figure 7). However, the majority of water usage is concentrated in Thalang District, and most of them are located in the same area as the operating EIA hotels with high water usage. If there is the development of water reservoirs, this area is also an option that should be considered in order to prevent long-distance transportation. Overall, the majority of the 178 hotels were concentrated in the western part of the province.

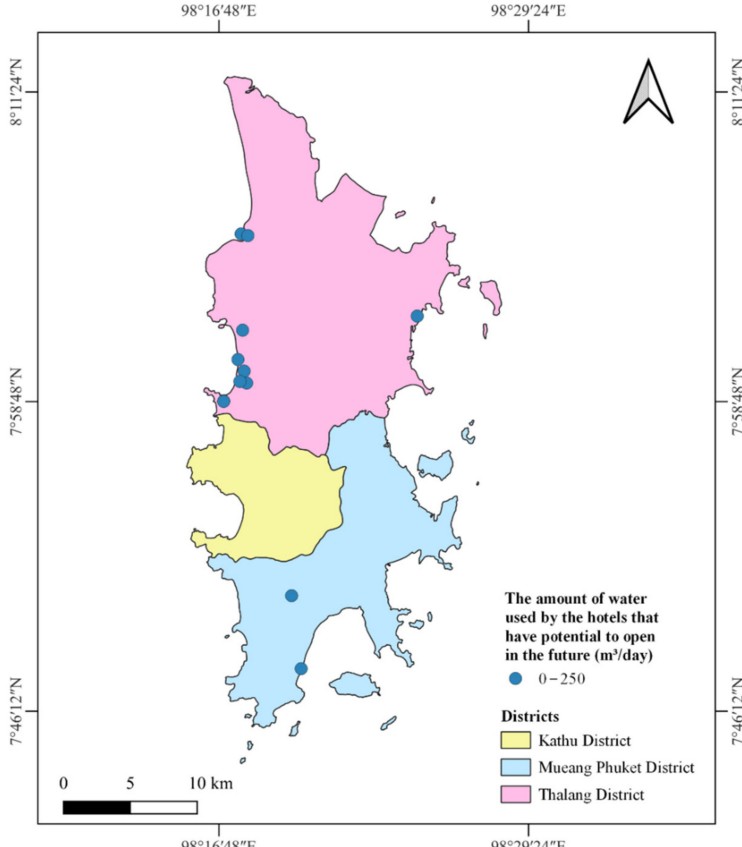

**Figure 7.** The locations of the hotels that require an Environmental Impact Assessment (EIA) with the potential to open in the future and their water consumption per day.

The analysis of the average water consumption per day of the EIA hotels with the potential to open in the future is presented as follows:

(1) Average water usage per room per day divided by hotel size (small, medium, and large) calculated from the total water usage of the hotel (including administrative activities and all other parts).

As shown in Figure 8, there are a total of 12 EIA hotels (with 1339 rooms as per Table 10) with the potential to operate in the future, and their average water usage is $1.0 \pm 0.2 \, m^3/\text{room/day}$. A middle-sized hotel has the highest water usage at $1.4 \, m^3/\text{room/day}$ with a total of 90 rooms while a hotel with the least water usage is at $0.7 \, m^3/\text{room/day}$ which is a large-sized hotel with a total of 219 rooms.

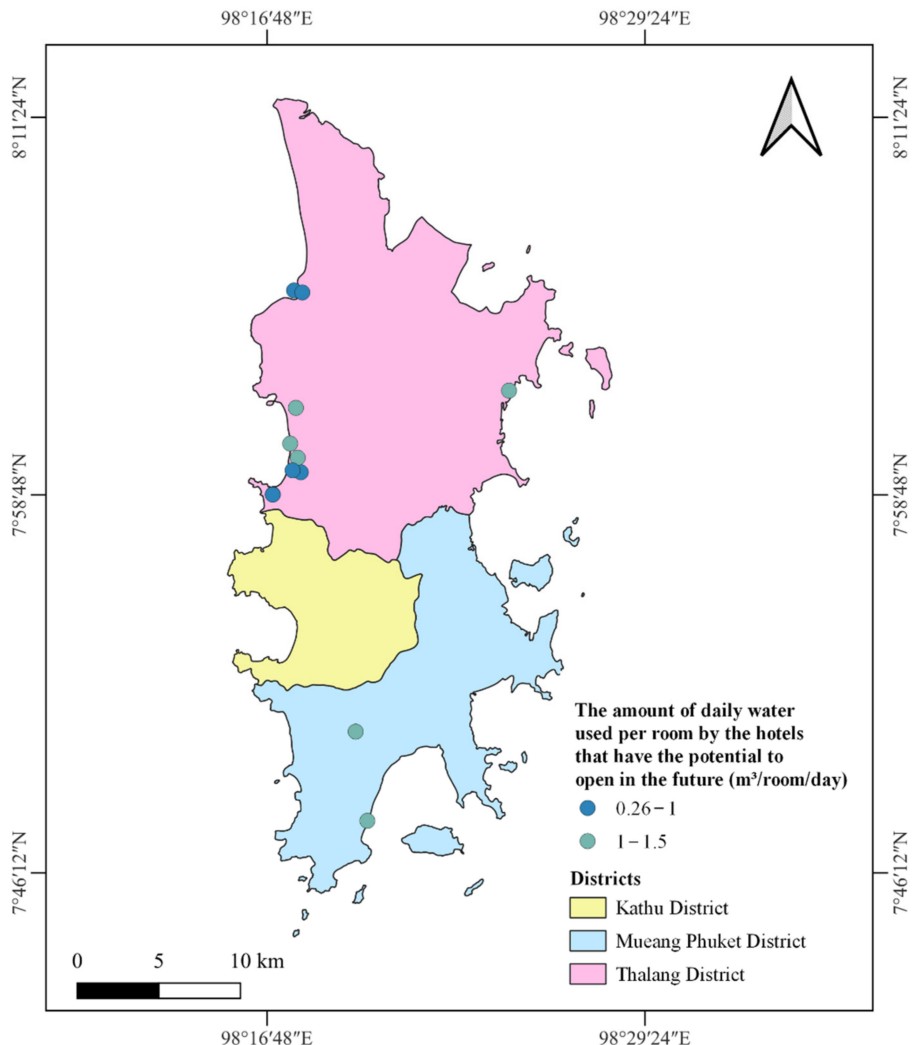

**Figure 8.** The locations of the hotels that require an Environmental Impact Assessment (EIA) with the potential to open in the future, and their water consumption per room per day.

**Table 10.** Average water usage per room per day, calculated from the hotels with the potential to operate in the future.

| Daily Water Consumption per Room (m³/Room/Day) | Number of Hotels | Total Rooms |
|---|---|---|
| 0–1.0 | 6 | 797 |
| 1.0–1.5 | 6 | 542 |
| Total | 12 | 1339 |

From calculating the average water use per room per day divided by hotel size as per Table 11, the average water use of medium-sized hotel has the highest value at $1.1 \pm 0.2$ m$^3$/room/day while the average water uses of large and small hotels have the values at $1.0 \pm 0.2$ m$^3$/room/day and $0.8 \pm 0.2$ m$^3$/room/day, respectively. Overall, the results in this part are consistent with operating EIA hotels that the average water use of medium-sized hotel has the highest value. In summary, medium and large hotels have higher average water usage than small hotels due to the variety of activities provided to guests. It is possible that large hotels are better equipped to implement effective water management measures. This results in the average water usage per room to be reduced. Although the average water use of large hotels is less than that of medium-sized hotels, it still uses more water than small hotels.

**Table 11.** Average water usage per room per day, divided by hotel size with the potential to operate in the future.

| Hotel Size | Number of Hotels | Total Rooms | Total Water Consumption (m$^3$/Day) | Water Use per Room (m$^3$/Room/Day) | | | |
|---|---|---|---|---|---|---|---|
| | | | | Ave. | SD | Max | Min |
| Small [1] | 2 | 78 | 80.9 | 1.0 | 0.2 | 1.2 | 0.9 |
| Medium [2] | 8 | 804 | 871.3 | 1.1 | 0.2 | 1.4 | 0.9 |
| Large [3] | 2 | 457 | 348.6 | 0.8 | 0.2 | 0.9 | 0.7 |
| Total | 12 | 1339 | 1300.9 | | | | |

[1] Small-sized hotels: hotels with less than 50 rooms. [2] Medium-sized hotels: hotels with 50–150 rooms. [3] Large-size hotels: hotels with more than 150 rooms.

(2) Average water usage per room per day divided by hotel location.

According to Table 12, the average water use of hotels located in Mueang District is slightly more at $1.1 \pm 0.1$ m$^3$/room/day, while that of hotels located in Thalang District Water is at $1.0 \pm 0.2$ m$^3$/room/day.

**Table 12.** Average water usage per room per day, divided by the districts where the hotels (with the potential to operate in the future) are located.

| District | Number of Hotels | Total Rooms | Total Water Consumption (m$^3$/Day) | Water Use per Room (m$^3$/Room/Day) | | | |
|---|---|---|---|---|---|---|---|
| | | | | Ave | SD | Max | Min |
| Mueang | 2 | 211 | 241.9 | 1.1 | 0.1 | 1.2 | 1.1 |
| Thalang | 10 | 1128 | 1058.9 | 1.0 | 0.2 | 1.4 | 0.7 |
| Total | 12 | 1339 | 1300.9 | | | | |

The average water usages per room per day for EIA hotels that are still in operation and EIA hotels with the potential to operate in the future are similar at 1127 L/room/day and 961 L/room/day. This is close to [48] that hotels in Bangkok province have an average water usage rate of 1210 L/room/day, with three important variables that affect the hotel's water usage level: (1) physical characteristics (hotel capacity, floor area, and hotel standard ranking {1–5 stars}), (2) facilities and leisure structures that require water (swimming pools, gardens, golf courses, spa facilities, laundry service, kitchens), and (3) business and environmental management model (hotel chain affiliation, number of employees, daily rate (price/room), and water-saving measures [30].

### 3.4. Simulation of the Water Consumption Situation in the Hotel Sector

Hotels' water usage in Phuket varies seasonally on quite different scales. Therefore, it is important for the simulations to reflect the hotel's minimum and maximum water use according to the months of the tourist season and outside the tourist season, which are presented as follows:

### 3.4.1. Daily Water Consumption

Daily water consumption has been simulated into two scenarios: the low season scenario, accounting for 60% of occupancy and water consumption in each hotel, and the high season scenario, representing 100% of occupancy and water consumption in each hotel. These percentages are based on the occupancy rate reported in the research project for water management in Phuket Province [36]. For the hotels currently in operation and those with the potential to open in the future, it was determined that during the low season, there will be a water demand of 24,275 $m^3$/day, and during the high season, there will be a total water usage of 40,457 $m^3$/day (Table 13).

**Table 13.** Simulation of the water consumption scenario in the hotel sector that requires an environmental impact assessment (EIA) in Phuket Province.

| Hotels That Require an Environmental Impact Assessment (EIA) in Phuket Province | The Water Consumption ($m^3$/Day) | |
| --- | --- | --- |
| | Low Season (60%) | High Season (100%) |
| Currently operating | 23,494 | 39,156 |
| Possibility of opening in the future | 781 | 1301 |
| Total | 24,275 | 40,457 |

### 3.4.2. Annual Water Consumption

Annual water consumption has been simulated into three scenarios: the low season scenario (60% of occupancy), the normal case scenario, and the high season scenario (100% of occupancy). These percentages are also based on the occupancy rate reported in the research project for water management in Phuket Province [36]. The calculation of annual water consumption for the normal case was conducted based on the months of the high season (December to February) with an occupancy rate of 100%. Then, for the low season (May to September), the water consumption was calculated with an occupancy rate of 60%, and for the regular periods of March to April and October to November, the rate was set at 80%. These seasonal rates are cited from the study conducted by [15]. According to the study results of 178 hotels, water consumption of 8,860,021 $m^3$/year, 11,303,606 $m^3$/year, and 14,766,699 $m^3$/year was found in the low season scenario, normal case scenario, and high season scenario, respectively (Figure 9).

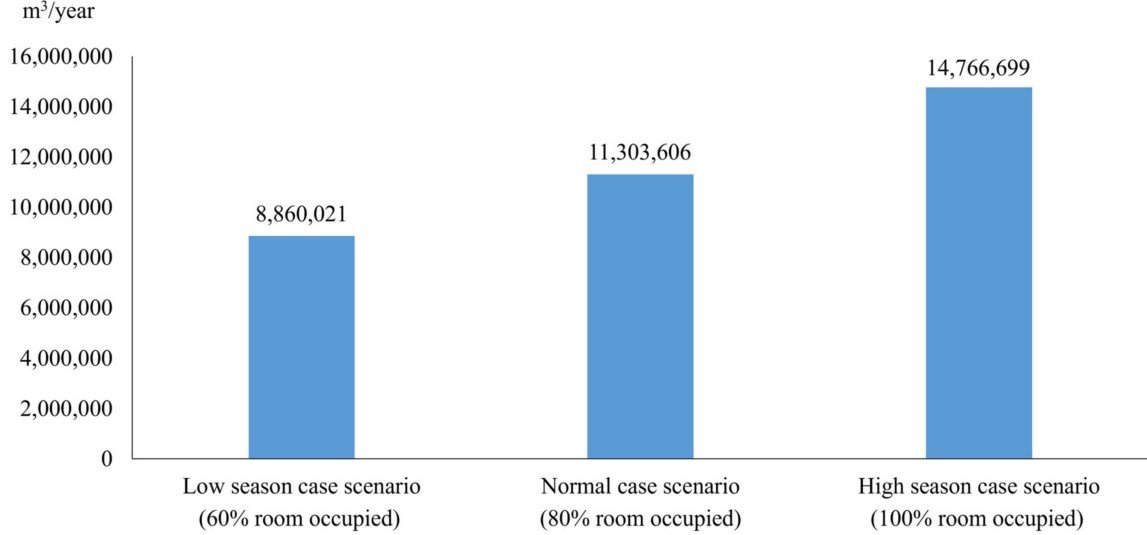

**Figure 9.** Annual water consumption simulation in the hotel sector ($m^3$/year).

*3.5. The Relationship between the Amount of Water Used in the EIA Hotels and the Amount of Water in Phuket*

Evaluation in the previous sections reveals that currently, Phuket Province still has a relatively high shortage of water. This section shows the significance of water storage and production by EIA hotels in Phuket Province themselves on the overall of the province. It can be divided into two cases: (1) EIA hotels do not have the capability to store and produce their own water, and (2) some EIA hotels have the capacity to store and produce their own water. A summary of the results from this study is given below.

3.5.1. Case 1: EIA Hotels Do Not Have the Capability to Store and Produce Their Own Water

In this case, the EIA hotels use 14,766,699 $m^3$/year of water, while Phuket's total water volume is 40,468,458 $m^3$/year. Consequently, water consumption in the EIA hotels of Phuket accounts for 36.5% of its total water volume (Figure 10).

**Figure 10.** The amount of water used in the EIA hotels and the total water supply in Phuket in a scenario where EIA hotels do not have the capacity to store and produce their own water.

3.5.2. Case 2: Some EIA Hotels Have the Capacity to Store and Produce Their Own Water

In this case, the EIA hotels use 14,766,699 $m^3$/year of water, while Phuket's total water volume is 43,056,162 $m^3$/year. Consequently, water consumption in Phuket's hotel sector is 34.3% of its total water volume (Figure 11).

As presented in Figures 10 and 11, the proportion of water consumption by EIA hotels to total water availability of Phuket province in Case Study 2 is less than in Case Study 1 at 2.2%. These values from 3.1–3.5 are for the estimation and analysis of the tourism carrying capacity in terms of water adequacy of Phuket in the next section.

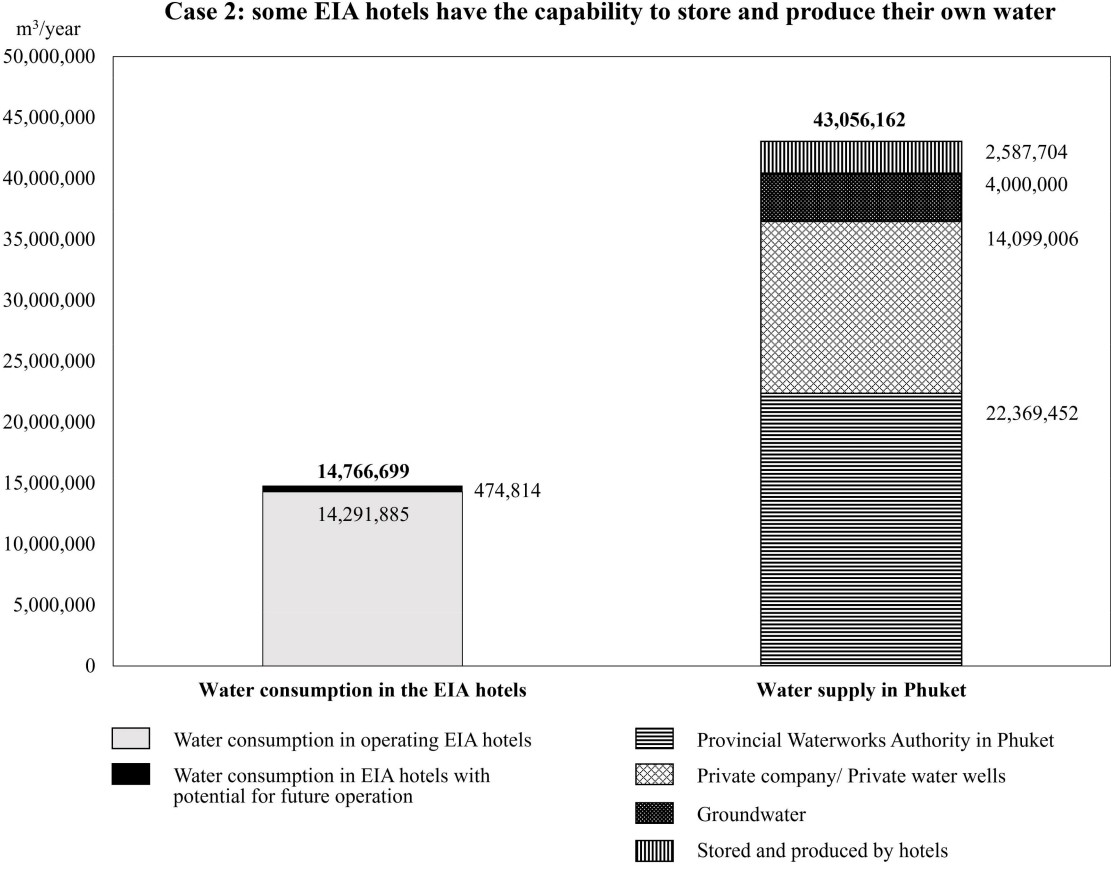

**Figure 11.** The amount of water used in the EIA hotels and the total water supply in Phuket in a scenario where some EIA hotels have the capacity to store and produce their own water.

### 3.6. Assessment of the Tourism Carrying Capacity in Terms of Water

Therefore, evaluating the tourism carrying capacity in terms of water adequacy is an important indicator that helps justify the suitable number of tourists that Phuket should limit in one year. It can also help indicate the importance of considering spatial potential and how much water retention efficiency should be accelerated or expanded to support tourism. The rate for calculating water usage by an ordinary tourist is at 350 L/person/day [41] as a minimum requirement, not at the rate of EIA hotels.

When evaluating the capacity to support water use by tourists in Phuket Province, it was found that for Case Study 1 hotels do not have the capability to store and produce their own water, Phuket Province has water left for consumption by tourists only 557,606 m$^3$/year or it can accommodate the maximum number of tourists at 1,593,160 people per year (calculated on the basis that each person comes for only one day). For Case Study 2, some hotels have the capacity to store and produce their own water; Phuket Province has water left for use by tourists at 3,145,310 m$^3$/year, or it can accommodate the maximum number of tourists at 8,986,600 people per year. It can be noticed that a value of only 2.2% from Section 3.5 results in an increase of seven million tourists because it is a percentage calculated at the provincial level. Nevertheless, from several years of statistical data between 2013 and 2019, Phuket Province has tourists in the range of 10–13 million people per year [49–51], while the Tourism Authority of Thailand—Phuket Office, which is an agency responsible for supporting tourism in Phuket Province set a target for the number of tourists for 2023 at 12 million people per year [52]. If the target is adhered to, after the use of the registered and latent population, Phuket water is insufficient to meet the needs of tourists in Phuket Province. Both Case studies 1 and 2 have a water shortage of 3,642,394 and 1,054,690 m$^3$/year, respectively.

In 2013, Phuket had a high number of tourists of 13 million [49], but it was able to pass that situation, delightfully, even though the amount of water should not be sufficient according to the results of this study. This is possibly due to the local use of self-drilled groundwater [15], which may be an amount that is beyond the 4,000,000 m$^3$/year of groundwater that the government estimates [40]. In addition, Phuket Province purchases water from Phang Nga Province (which results in increased water consumption costs). Phang Nga Province is likely to experience future economic development by having its own international airport. This situation could potentially result in increased water demand and the inability to share water with Phuket Province [53]. Moreover, there is a need to use water for other sectors such as agricultural activities and downstream ecosystems. Therefore, it is necessary to accelerate the development or expansion of water storage areas to increase tourism carrying capacity and to serve all water demands.

### 3.7. Guidelines for Augmenting the Water Supply in Phuket Province

Currently, the issue of water scarcity in Phuket continues to be a pressing concern that requires immediate attention. The problem of water loss in waterworks production can be addressed in the short term, and relevant agencies should seek solutions to manage these issues and increase the water supply to the province. Additionally, there is a need to enhance the region's capacity to harvest rainwater to meet the demands of all sectors. Rainfall data spanning the past 15 years (from 2008 to 2022) indicates a slightly continuous upward trend in rainfall, although precipitation levels may vary from month to month (Figure 12). Therefore, by identifying the primary challenge of water scarcity in Phuket Province, better management of available water resources should be achieved.

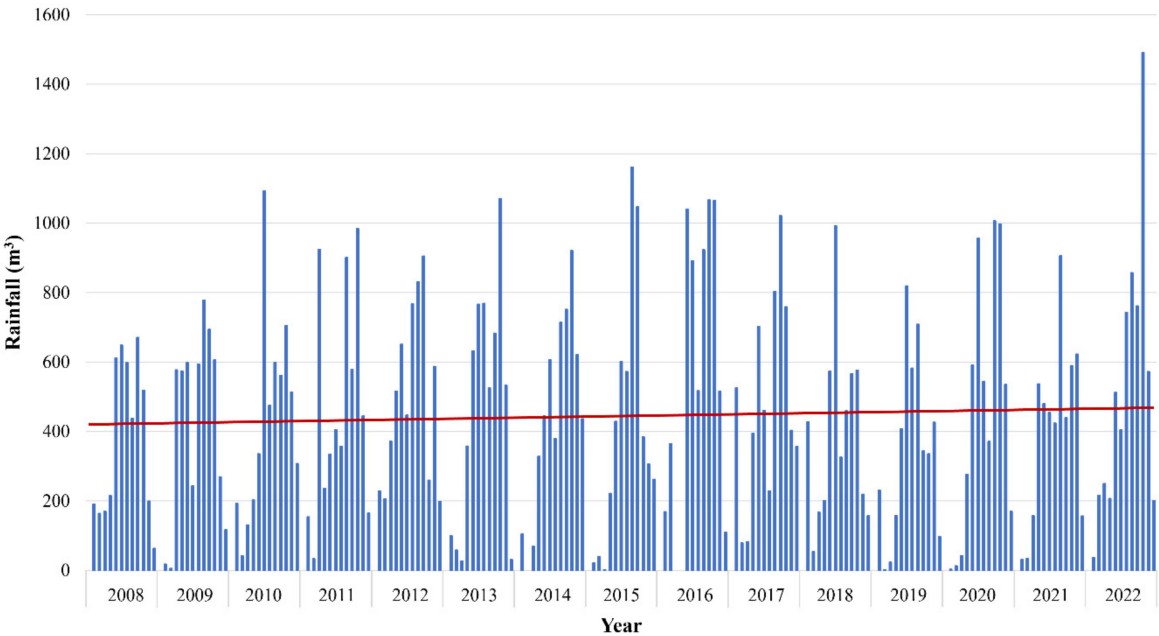

**Figure 12.** The rainfall data for the past 15 years (2008 to 2022). Note: This graph shows rainfall data for Phuket from only two stations: Station codes 564,201 and 564,202. The blue bars show monthly rainfall and the red line shows the trend of changes in rainfall over 15 years.

The main problems contributing to water shortages in Phuket Province include the limited water storage capacity of reservoirs and the minimal rainfall during the dry season.

Even though efforts are being made to solve the problem by expanding reservoir areas and other water bodies, more than doubled from 2000 to 2018, transformed many types of areas such as agricultural lands, forests, or even urban & built-up [49,54]. These issues require more significant attention and action within the administrative sector. Addressing the problem of water scarcity in Phuket Province can be substantially achieved through

integrated water management, which necessitates collaboration among all sectors, including government, private sector, and the public. In order to ensure an ample water supply for all sectors, the key authorities in the province have formulated water management policies. These policies aim to establish sustainable water management practices that can positively impact the residents' quality of life, enhance the province's capacity to accommodate tourists and stimulate the local economy in Phuket Province. At present, the Phuket Provincial Administrative Organization has determined the following solutions to solve the water shortage problem: (1) increase water storage in small areas by developing small irrigation projects and renting the old tin mines, (2) improve the water storage capacity of the existing main reservoirs, (3) increase the amount of water in the main reservoirs, (4) develop drainage systems between the main reservoirs, (5) develop groundwater pumping systems, and (6) buy and sell water from outside Phuket province [16].

This study also indicates to us that many hotels are important storage and producers of clean water in the province, which may happen because of the favorable nature of the area. Yet, supporting business sectors such as hotels to prepare their own water sources for emergency use is another possible option. Another important suggestion is to improve the province's water pipeline system, which regularly wastes around 30% of clean water, calculated from the difference in distribution and sold volumes found in the data of this study (the year 2017) and consistent with current data (the year 2023) [35]. This may require a huge budget, but if the amount of water lost (780,000 $m^3$/month or 9,400,000 $m^3$/year [35]) can be taken back into the pipe system and used again, It will be able to accommodate an additional 26,857,143 tourists/year, which is sufficient to meet the province's tourist-target.

## 4. Conclusions

The study identified 178 hotels in Phuket Province subject to EIA, including operating hotels and potential future ones. Collectively, these hotels consumed 14,766,699 $m^3$/year of water, with individual consumption influenced by factors like room size and on-site activities. The western part of Phuket had a higher concentration of hotels, resulting in significantly greater water usage.

EIA hotels in Phuket have made an effort to reduce water shortages in the province by storing and producing water for their own use, with a water amount of 2,587,704 $m^3$/year. Two scenarios were analyzed: EIA hotels with and without their own water storage and production. In the first and second scenarios, the EIA hotels used 36.5% and 34.3% of Phuket's total water amount, respectively. However, an assessment of tourism carrying capacity discloses that Phuket has insufficient water to meet the needs of tourists. Taking into account all water requirements of the registered and latent population, it becomes clear that Phuket Province must procure a minimum of 3,642,394 $m^3$ and 1,054,690 $m^3$ of water annually to meet the tourism target of 12 million people per year for both Case Studies 1 and 2 while this does not include demands from other sectors such as agriculture and maintaining downstream ecosystems. Water scarcity in Phuket remains a pressing concern, requiring prompt attention and effective solutions from water management agencies.

Although this study was not based on data directly collected from hotels' operations, data received from the EIA reports also give an approximate view of water use at the provincial level. Obtaining water usage data from hotels is notoriously difficult. This study may serve as a guideline for those involved in the area's water management to use as an example in estimating the level of water use in their own area.

Hotels with a large number of rooms and large sizes have a high water consumption. This is because medium- and large-sized hotels often offer additional amenities to their customers. As a result, water usage activities are often higher than in small hotels, but of course, hotel room rates also tend to vary according to size and services. The eco-efficiency study between average room rental rate and the amount of water required in the overall province is another interesting issue for further study in considering the worthiness of the size and service of the hotel, water required to serve, and the income that the province can receive which is linked to the policy to support or control the size of the hotel.

**Author Contributions:** Conceptualization, T.S. and K.P.; methodology, T.S., K.P., T.K., T.A., P.P. and C.P.; project administration, K.P.; software, T.K., T.A. and P.P.; validation, T.S., K.P. and C.P.; formal analysis, T.S., K.P., T.K., T.A., P.P. and C.P.; investigation, K.P., T.K., T.A. and P.P.; resources, K.P.; data curation, T.S., K.P. and C.P.; writing—original draft preparation, T.S. and K.P.; writing—review and editing, T.S., K.P. and C.P.; visualization, T.S., T.K., T.A. and P.P.; supervision, K.P. All authors have read and agreed to the published version of the manuscript.

**Funding:** The authors are grateful to the sponsorship from the National Research Council of Thailand (NRCT) and the National Research Foundation of Korea (NRF) Scientist Exchange Program 2021 for providing a chance to obtain useful advice from Taehyeung Kim during the program.

**Institutional Review Board Statement:** Not applicable.

**Informed Consent Statement:** Not applicable.

**Data Availability Statement:** Data are contained within the article.

**Acknowledgments:** The authors express their gratitude to the geo-informatics facilities and services of the Faculty of Environment and Resource Studies, Mahidol University, for geo-informatics software installation and usage consultation. The authors also extend their thanks to Sirasit Vongvassana for his technical support on GIS, to Penprapa Suwimolsatian for the advice on acquiring information on hotels that passed an Environmental Impact Assessment (EIA), to Surachet Pinkaew for suggesting sources of information pertaining to this research, and to Suthathong Homya for the initial idea of this project. Lastly, the authors acknowledge the Thai Meteorological Department for providing monthly rainfall data for the past 15 years.

**Conflicts of Interest:** The authors declare no conflict of interest.

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
