# Peer review of "The Connection between Phuket’s Water Supply and the Hotel Sector’s Water Use for Assessment of Tourism Carrying Capacity"

_sustainability, doi:10.3390/su16020621_

Round 1
Reviewer 1 Report
Comments and Suggestions for Authors
Dear Authors,
I have reviewed your paper and found it to be relevant and generally aligned with scientific writing norms. However, I recommend some minor revisions to enhance its overall quality. Below, I have outlined my observations and recommendations:
Abstract: The abstract is well-structured. However, I encourage the authors to provide a brief description of the methodology used and highlight the key findings.
Introduction: The introduction should be more robust and well-justified. I recommend that the authors clearly articulate the research gap they intend to address and state the paper's structure.
Literature Review: I suggest extending the LR by referencing relevant previous studies for more comprehension.
Methodology: While the research method is relevant to the paper's objectives, it lacks sufficient description. I encourage the authors to justify the techniques and approaches used and explain their suitability within the research context.
Results: The results are well-presented. However, a discussion section is missing. I recommend including a discussion that relates to the obtained results, with necessary comparisons.
Conclusion: The conclusion should summarize the main contributions of the research in advancing knowledge in the field and highlight the main limitations.
References: There appear to be too few references for such a hot topic. I suggest adding more pertinent and recent papers to enhance the overall quality of the paper. (Please refer to sustainability author guidelines for the suitable format).
Reviewer 2 Report
Comments and Suggestions for Authors
Dear authors,
your mission is clear, Phuket hotels use too much water. I find the message clear, but the paper morepractica than academic. I would advice to change the lay-out of fig 7-9, the graphs are a bit difficult to read: height of column is easier to read then area. Furthermore, you may want to use another measure to compare hotels, e.g. water use per room. That would make it easier to compare hotels of different sizes.
Reviewer 3 Report
Comments and Suggestions for Authors
As shown in the attached file.

Reviewer 4 Report
Comments and Suggestions for Authors
Comment 1: The clarity of the paper illustrations can be further improved. Compared with other drawings, the clarity of Figure 1 is poorï¼›
Comment 2: Add the introduction of research Methods in section 2. Materials and Methodsï¼›
Comment 3: The classification of case1 and case2 in Section 3.4 is not necessary, the information on case2 covers the information on case1ï¼›
Comment 4: The overall research method of this paper is too simple, and the research conclusion focuses on the analysis and summary of the current situation. Future prediction can be further carried out to enrich the research content.
Comments on the Quality of English LanguageThere is still room for optimization of the writing logic of the article. In addition, the language expression of individual chapters is a little verbose and can be further simplified
Round 2
Reviewer 4 Report
Comments and Suggestions for Authors
This paper investigates the connection between water supply in Phuket and the water use in the hotel sector for tourism carrying capacity assessment. The study utilizes geographic information system (GIS) to analyze the spatial distribution of water use in hotels and the available water in Phuket. The findings reveal that the amount of water in the province is insufficient to meet the needs of tourists, and urgent measures are needed to increase the available water. The study is meaningful and I am interested in it. However, there are still some points need to be improved:
(1) Concept ambiguity in Line 100: ‘the rainy season from April to November and the summer season from December to March’. It is recommended to check the distinguishability between "rainy season" and "summer season". Should the term "summer season" in this context be "dry season"
(2) Table 1 formation. There is no gap between the second row and the third row in the ‘Type of data’ column, which might lead to misunderstanding in this case. It is recommended that a post-segment distance be added.
(3) It is recommended that XT = Total water amount in Phuket province (m3/year), which may be better to be well-understand..
(4) It is recommended that a further explanation is needed to tell whether the count of hotels cooperating with ‘The Provincial Waterworks Authority and private companies’ includes the hotels only cooperating with ‘The Provincial Waterworks Authority’.
(5) In the average water uses analysis of the hotels with potential to operate in the future, medium-sized hotels were found to be the most wasteful of water, and the average water uses of large hotels were significantly smaller than medium-sized. It is recommended that the reasons for this phenomenon be briefly analyzed.
(6) In Section 3.2, in general, population size can be a factor in judging the level of water use in an area, but it does not provide a comprehensive picture. Where possible, it is recommended that a combination of factors affecting water use should be considered.
(7) Grammar and sentence format issues. For example, Line 597,’(780 ,000 m3/month)’.
Comments on the Quality of English LanguageThe paper demonstrates a good command of English language and academic writing conventions. With some refinements in sentence structure, readability, and clarity, it can be further improved for its academic audience. For instance, some sentences are lengthy and packed with information, which can be challenging for readers to follow. Breaking down complex ideas into simpler sentences could enhance readability.
Round 3
Reviewer 4 Report
Comments and Suggestions for Authors
This paper investigates the connection between water supply in Phuket and the water use in the hotel sector for tourism carrying capacity assessment. The study utilizes geographic information system (GIS) to analyze the spatial distribution of water use in hotels and the available water in Phuket. The findings reveal that the amount of water in the province is insufficient to meet the needs of tourists, and urgent measures are needed to increase the available water. The paper has been well-revised. However, there are still a few points need to be improved before publication:
(1) In Lines 17-20, most numbers are listed without units, there are too many ‘,’ among them, which is hard to read. It is recommended that add the unit after each number.
(2) It is recommended that the figure format be uniform, especially Figure 4 to Figure 8. For example, given that the overall area covered by the study remains the same, it is desirable to maintain consistency in latitude and longitude across the maps. Figure 3 and Figure 4 in the text differ in their longitude scales, which may affect the aesthetics. In addition, the resolution of Figure 12 is a bit low, and some of the information is presented in a blurred manner.
(3) A few grammatical issues need to be revised. For example, Line 584, there is a spare ‘.’ before citation [16]. In addition, Line 592, there is a spare ‘(’.
Comments on the Quality of English LanguageThe paper demonstrates a good command of the English language and academic writing conventions. With some refinements in sentence structure, readability, and clarity, it can be further improved for its academic audience.
